# Metabolic Signaling Cascades Prompted by Glutaminolysis in Cancer

**DOI:** 10.3390/cancers12092624

**Published:** 2020-09-14

**Authors:** Raj Shah, Suzie Chen

**Affiliations:** 1Susan Lehman Cullman Laboratory for Cancer Research, Ernest Mario School of Pharmacy, Department of Chemical Biology, Rutgers University, Piscataway, NJ 08901, USA; rajs@scarletmail.rutgers.edu; 2Joint Graduate Program in Toxicology, Rutgers University, Piscataway, NJ 08901, USA; 3Rutgers Cancer Institute of New Jersey, New Brunswick, NJ 08901, USA

**Keywords:** metabolism, cancer, glutamine, glutamate, glutaminase (GLS), glutaminolysis, CB-839

## Abstract

**Simple Summary:**

Within the last few years, accumulating evidences suggest the involvement of altered metabolisms in human diseases including cancer. Metabolism is defined as the sum of biochemical processes in living organisms that produce and consume energy. Tumor growth requires restructuring of cellular metabolism to meet the increasing demand for building blocks to support the ever-increasing cancer cell numbers. The principle of perturbed metabolism in tumors is known for 50–60 years, it regains greater appreciation within the last few years with the realization that there is interdependency between metabolism and all aspects of cellular function including regulation and control of cell growth. Tumor cells do not need stimulation signals from the surrounding environment to promote cell proliferation; in some cases, the tumor cells can generate their own growth signals. In order to support the continuous tumor cell growth even under stressful conditions, a change in metabolism is necessary to fulfill the continuous demand for energy and building blocks. A better understanding of the relationship between tumor environment and altered cell metabolisms will provide valuable insights to design innovative approaches to limit the supply of energy and macromolecules for the treatment of cancer including melanoma.

**Abstract:**

Aberrant glutamatergic signaling has been implicated in altered metabolic activity and the demand to synthesize biomass in several types of cancer including melanoma. In the last decade, there has been a significant contribution to our understanding of metabolic pathways. An increasing number of studies are now emphasizing the importance of glutamate functioning as a signaling molecule and a building block for cancer progression. To that end, our group has previously illustrated the role of glutamatergic signaling mediated by metabotropic glutamate receptor 1 (GRM1) in neoplastic transformation of melanocytes in vitro and spontaneous development of metastatic melanoma in vivo. Glutamate, the natural ligand of GRM1, is one of the most abundant amino acids in humans and the predominant excitatory neurotransmitter in the central nervous system. Elevated levels of glutaminolytic mitochondrial tricarboxylic acid (TCA) cycle intermediates, especially glutamate, have been reported in numerous cancer cells. Herein, we highlight and critically review metabolic bottlenecks that are prevalent during tumor evolution along with therapeutic implications of limiting glutamate bioavailability in tumors.

## 1. Introduction

Hanahan and Weinberg first proposed the six biological hallmarks of cancer in 2000, and four additional new hallmarks as necessary traits during the development and progression of cancer were subsequently added in 2011 [1]. These hallmarks are unregulated cell growth, anti-apoptosis signals, induction of angiogenesis, unresponsiveness to growth suppressors, metastatic capabilities, replicative immortality, genomic instability, immune system evasion, tumor-specific inflammatory response, and transformation of cellular metabolism. Cancer cells are constantly adapting to the hosts’ defense by manipulating intrinsic and extrinsic biological pathways. Within the last two decades, reprogramming of energy metabolism has emerged as a popular and valuable therapeutic cancer target to study. Cell metabolism, simply defined, is the set of complex biochemical processes occurring in a cell required to sustain life. Due to their utterly plastic nature, cancer cells may utilize a plethora of pathways for energy production [2]. Metabolic pathways are composed of numerous steps that are highly regulated, and it is possible for metabolites formed in one pathway to feed into other biosynthetic pathways. In cancer, these pathways differ depending on the tissue of origin and are often rewired allowing tumor cells to sustain hyper-growth and proliferative states.

## 2. Aerobic Glycolysis “Warburg Effect”

Cancer cells employ a different metabolic strategy than normal cells to satisfy their energy requirements and sustain cellular proliferation. Under aerobic conditions, normal cells acquire their energy primarily from the conversion of glucose to pyruvate by a process known as glycolysis, which occurs in the cytosol. The pyruvate then enters the tricarboxylic acid (TCA) cycle where it converts into carbon dioxide in the mitochondria via oxygen-consuming cellular respiration [3,4]. However, under hypoxic conditions where oxygen is not readily available, cells prefer to rely more on anaerobic glycolysis, which converts glucose into lactate instead of pyruvate, resulting in decreased availability of pyruvate for mitochondrial respiration (oxidative phosphorylation). It has been noted, however, that cancer cells often produce large amounts of lactate regardless of the availability of oxygen, and this form of metabolism is referred to as “aerobic glycolysis” or the “Warburg effect” [3,5,6]. This phenomenon was first observed in 1924 by the Nobel laureate and German scientist, Otto Warburg. Moreover, a recent review has highlighted the roles of hypoxia inducible factor 1 (HIF-1) and the PI3K/AKT/mTOR pathway in reprogramming metabolism of cancer cells towards preferential utilization of aerobic glycolysis as an energy source [7]. While aerobic glycolysis is considerably less efficient than cellular respiration in terms of adenosine triphosphate (ATP) generation—2 ATP versus 36 ATP, respectively—when the cell microenvironment is nutrient limited, aerobic glycolysis can provide an advantage for tumor-cell growth by generating ATP at a faster rate [8]. It has been suggested, however, that the reason for this ‘metabolic switch’ is not to increase ATP production, since the amount of ATP in a proliferating cell is not significantly different from a resting cell, but rather to provide the building blocks for macromolecular synthesis [9,10].

## 3. Tricarboxylic Acid (TCA) Cycle

Under aerobic conditions, pyruvate, the end product of glycolysis, enters the mitochondria to be oxidized to acetyl Coenzyme A (CoA), which combines with oxaloacetate to start the TCA cycle and oxidative phosphorylation [2]. One predominant metabolic rewiring activity distressing the TCA cycle is that many cancer cells exhibit remarkable dependence on glutamine and cannot survive with glutamine deprivation [11]. This phenomenon is often referred to as ‘glutamine addiction’. Experimental evidence suggests that glutamine is the major respiratory fuel for energy production in tumor cells [12]. Glutamine is the most abundant amino acid in human blood [13,14]. In addition to being a nitrogen donor for protein and nucleotide synthesis, glutamine provides for anaplerosis to refill the mitochondrial carbon pool. During periods of rapid growth, the demand for glutamine surpasses its supply in many cancer cells [15]. It has been demonstrated that tumor cells can utilize glutamine for citrate production through the reversal (reductive carboxylation) of the TCA cycle [16]. First, glutamine is de-aminated to glutamate, via glutaminase (GLS), which is then converted to α-ketoglutarate. Next, α-ketoglutarate undergoes reductive carboxylation to generate isocitrate by isocitrate dehydrogenase (IDH). At last, isocitrate is catalyzed by aconitase to produce citrate, which is converted to acetyl CoA by ATP citrate lyase [17]. Overall, both glutamine and glucose may provide the carbon skeletons and co-factors, such as NADPH and ATP, for cancer growth and survival.

## 4. Factors that Can Potentially Affect Metabolic Activity in Cancer Cells

Earlier, only genomic modifications that result in the activation of oncogenes, loss of tumor suppressors, or mitochondrial DNA mutations were expected to regulate cancer-cell metabolism. Lately, it has been recognized that the metabolic phenotype of cancer cells can also be influenced by several non-genetic factors. As the number of neoplastic cells increases in the tumor, nutrient and oxygen availability gradually begins to decrease. This triggers the formation and growth of new blood vessels that are poorly formed and inefficient [18]. Subsequent changes in the availability of nutrients are known to have a significant impact on actively proliferating carcinomas. Moreover, contrary to the traditional view that cells can take up and utilize nutrients whenever their reserves are depleted, nutrient uptake is strictly regulated by growth factor signaling [19]. In addition to nutrient availability, metabolism could also be modulated by the surrounding tumor microenvironment (TME) of the cancer cell. Hypoxic conditions in the TME could activate a transcriptional program that could theoretically change the metabolic profile of cancer cells [20,21]. There have been reports suggesting that the molecular basis for the shift from oxidative to reductive glutamine metabolism in mammalian cells is linked to HIF-1α activity [22]. An area that has distorted the viewpoints of multiple experts is the questionable contribution of stromal cell-generated metabolites to the tumor and whether these metabolites promote or inhibit tumor advancement. Taken together, these insights shed light on intrinsic and extrinsic factors that disrupt metabolism, all of which can have important implications in cancer development and progression.

## 5. Physiological Role of Glutamate in Normal and Cancer Cells

Glutamate is the most abundant and multifaceted biomolecule that plays a fundamental role in multiple metabolic processes and signaling in human cells. Glutamate, the predominant excitatory neurotransmitter in the central nervous system (CNS), is also involved in several non-neuronal cellular functions through interaction with different receptors [23]. Glutamate has been shown to regulate proliferation, migration, and survival of neuronal precursor cells during development of the brain [24]. Glutathione (GSH), an important scavenger of reactive oxygen species (ROS) found in the cytosol of all human cells, is made up of glutamate, glycine, and cysteine. Tumor cells express elevated levels of antioxidant proteins, such as GSH for detoxification [25], further endorsing the importance of glutamate. Post conversion to α-ketoglutarate by glutamate dehydrogenase (GLDH), glutamate—produced from oxidation of glutamine—may enter the TCA cycle to supply intermediates for cell growth [2]. When the supply of glutamine is scarce, glutamate, and ammonia can be condensed in an ATP-dependent manner for de novo glutamine synthesis by glutamine synthetase (GS), otherwise known as the glutamate–ammonia ligase (GLUL) [26]. Most metabolic pathways where the free ammonia is utilized, the efficiency of nitrogen utilization is maximized as the cells prefer to transfer nitrogen from amino acids [27,28]. However, the GS catalyzed reaction is special, because it plays an important role in nitrogen metabolism, ammonia detoxification, and cell signaling [29]. Evidence points to the involvement of glutamate in cancer progression and regulation of the TME [30]. Moreover, supplementation of glutamic acid in conditioned media stimulated proliferation in slow-growing melanoma cells [31,32], indicative of a growth advantage. This likely has to be attributed to the fact that abundant glutamate in the TME supports efficient carbon utilization for anabolism and growth [33]. In fact, studies have showed that excessive glutamate concentrations in the TME of glioblastoma patients results in accelerated tumor growth [34] possibly leading to epileptic seizures in those patients [35]. Furthermore, glutamate antagonists have been shown to limit tumor growth, migration, and invasiveness in human tumors, including breast, colon, lung, and astrocytoma, showing their anticancer potentials [36]. Increasing knowledge of glutamate signaling in tumorigenesis may lead us towards finding putative targets against various components of glutamate-mediated signaling. Interestingly, one of the most noticeable reprogramming events in cancer-cell metabolisms is the preferential utilization of glutamate for reductive metabolism even under normoxic conditions. Recently, several reports have linked resistance to serine/threonine protein kinase BRAF (BRAF) inhibitors with augmented glutamine dependency [37,38,39], suggesting that altered glutamate-dependent anabolic pathways may be central to acquiring drug resistance in cancers including melanoma. Additionally, two large omics studies have highlighted the role of glutamate-mediated activation of G-protein coupled receptors (GPCRs) in conferring BRAF inhibitor resistance in melanoma cells [40,41].

## 6. Glutaminolysis

In the late 1950s, it was found that some cancer cells could not survive without the addition of exogenous glutamine in the growth media, suggesting that tumor cells are highly dependent on glutamine for survival and growth [42]. Experimental evidence shows that glutamine is the major respiratory fuel for energy production in tumor cells [43]. The ability of glutamine to satisfy the bioenergetic needs and provide intermediates for macromolecular synthesis required for cell growth is important in tumor-cell metabolism [12]. Thus, the metabolism of glutamine is considered another important hallmark besides the “Warburg effect” in tumor-cell metabolism. In humans, glutamine has the highest concentration in the blood plasma relative to other amino acids, ranging from concentrations of 0.5 to 1 mM [10,15]. Due to its extracellular abundance, glutamine is transported into the cell via the SLC1A5 (ASCT2) transporter [44,45,46]. The internalized glutamine is then oxidized through the loss of its amide group to form glutamate, by a mitochondrial-associated enzyme called glutaminase (GLS) [47,48]. GLS is an amidohydrolase that is often referred to as the “key gatekeeper” of glutamate-driven glutaminolysis [49]. The reverse reaction is catalyzed by another enzyme, glutamine synthetase (GS), which catalyzes the conversion of glutamate back into glutamine, and has been implicated in cancers, such as primary liver cancer and hepatocellular carcinoma [50,51]. Byproducts of the “glutaminase” reaction are used for synthesis of purines, pyrimidines, NAD^+^ cofactors, amino-sugars, glutathione, and non-essential amino acids (NEAA), such as alanine, asparagine, and phosphoserine [45,46,52].

The human genome encodes two distinct isoforms of glutaminases: kidney-type glutaminase (KGA) and liver-type glutaminase (LGA). Different isoforms of each enzyme arise from alternative splicing and surrogate promoter mechanisms [53]. KGA, which has ubiquitous distribution, is encoded by the *GLS1* gene on chromosome 2, whereas LGA, mainly expressed in liver tissues, is derived from the *GLS2* gene on chromosome 12. KGA exists as two splice variants through alternative splicing: one expressing the full length form of the *GLS1* gene, which retains the acronym KGA, and the other is termed as kidney glutaminase isoform C (GAC), which has a 71 residue shorter carboxy-terminus [49]. Numerous evidence implicates that upregulation of KGA, especially GAC (jointly referred to as GLS henceforth), plays a critical role in tumor proliferation throughout various types of cancers, such as glioma, lymphoma, non-small cell lung cancer, prostate cancer, and triple-negative breast cancer [54,55,56,57]. Furthermore, downregulation or inhibition of GLS has slowed the proliferation of these tumor cells [57,58]. GLS inhibition has been shown to enhance the effectiveness of chemotherapy [59] and also improve the efficacy of other targeted therapies [60,61], suggesting the critical role of targeting GLS in an attempt to improve overall patient response. Elevated GLS levels are functionally linked to the oncogenic transcription factor, Myc. Myc-induced cell growth [62] has emerged as an important player in numerous cancer types [54]. The vital role of glutamate in cancer-cell proliferation suggests that glutaminolytic enzymes could be attractive targets for therapy. A schematic illustration describing the metabolic fates of glutamate is shown in Figure 1.

## 7. Cancer Cells Amplify the Release of Extracellular Glutamate

The role of glutamatergic signaling in tumor biology has been increasingly studied in a variety of malignancies including neuronal tumors, melanoma, breast cancer, prostate cancer, etc. Melanoma cells release excess glutamate into the extracellular environment to warrant constitutive activation of the GRM1 receptor [63]. Moreover, several later studies conducted in different cancer models supported these findings when they detected a more than threefold increase in extracellular glutamate from GRM1 expressing cells compared with controls [64,65,66]. Similar to the activation of metabotropic glutamate receptors, enhanced glutamate release can also lead to stimulation of ionotropic glutamate receptors (iGluRs). Studies on melanocytes and associated tumors have shown that iGluRs modulate microphthalmia-associated transcription factor (MITF), a factor responsible for melanocyte lineage commitment, and treatment with AMPA receptor antagonists reduces MITF levels, reduces migration, and induces apoptosis [67]. In glioblastoma, calcium entry caused by glutamate-mediated activation of the AMPA receptor increases phosphorylation of cell proliferation and survival pathways [68]. AMPA receptor activation contributes to the lower mitogenic threshold required for oncogene induced signaling and transformation in early pancreatic cancer [69].

Briggs et al. proposed that large amounts of extracellular glutamate, secreted by triple-negative breast cancers, has the potential to inhibit cystine uptake by the cystine-glutamate antiporter (xCT) system [70]. This intracellular depletion of cysteine can increase HIF-1α expression due to the inactivation of the main HIF-1α prolyl-hydroxylase [70]. HIF-1α prolyl-hydroxylases are responsible for the degradation of HIF-1α. Others have reported that the molecular basis for the rewiring of anabolic glutamate metabolism in mammalian cells is linked to HIF-1α activity [22]. HIF-1α could also be activated by the PI3K/AKT/mTOR signaling pathway [71], which is upregulated in numerous cancers including GRM1-expressing melanoma cells.

It is well known that stem cells of the neural crest give rise to the cells of the central nervous system (CNS), including astrocytes, glia, and neurons [72]. Melanocytes of the skin also arise from the neural crest stem cells. Due to the similar progenitor origin of the CNS cells and melanocytes, Prickett and Samuels proposed that they may share similar signaling pathways important for homeostasis, proliferation, growth, and overall survival [73]. Glioma, a cancer arising from glia cells in the brain, uses glutamate as an autocrine or paracrine signal to promote cellular migration and invasion [74]. Results from a recent study by Pei et al. indicate that glutamatergic signaling may provide positive feedback through metabolic reprogramming and genetic switching to accelerate glioma duplication and progression [75]. Glioma cells release excess glutamate through the xCT antiporter, which causes the excitotoxic death of neurons and permits tumor-cell expansion [76,77]. That evidence that glutamate-secreting glioma cells exhibit a distinct growth advantage is also noteworthy [78]. It was previously reported that the brain is a preferred site for a secondary melanoma tumor to arise once it becomes metastatic [79]. Therefore, it is interesting to note that when this occurs, excess glutamate released by melanoma cells may further promote tumor growth in a similar fashion to glioma [63]. One of the possible ways for these cells to obtain enough glutamate for subsequent release is by elevating the consumption of glutamine into cells followed by conversion to glutamate via GLS. In addition, enhanced glutamate release has been observed in melanoma as well as breast cancer and prostate cancer cell lines, further supporting the importance of glutamatergic signaling in several malignant phenotype [30].

## 8. Can Glutamate Be Used as a Prognostic Biomarker?

The identification of a reliable predictive clinical biomarker is crucial for precision medicine. Predictive biomarkers are biological molecules detected in most patients and are frequently correlated with treatment responses [80]. Personalized/precision medicine is the future for human disease treatments, and it is essential to identify clinically relevant biomarkers, which can be easily applied in the clinic. Most pre-clinical cancer studies only assess for the efficacy of drug(s) on tumor progression, but it is crucial to also identify predictive biomarkers for treatment responses. Identification of these biomarkers will give clinicians opportunities to make suitable and rational decisions in therapeutic options.

A prognostic tool that has recently been developed measures glutamine addiction in patients [46]. First, a patient is injected with radioactive ^18^F-labeled 2S, 4R stereoisomer of 4-fluoroglutamine (^18^F-glutamine), followed by a position emission tomography/computed tomography (^18^F-glutamine-PET/CT) scans, in contrast to the conventional ^18^F-glucose (FDG-PET/CT) scan, which measures the Warburg effect [46,81,82]. ^18^F-glutamine-PET/CT scans are useful in clinics to stage cancer, assess treatment responses, and predict the prognosis of the disease [82]. The development of this tool was only possible due to the understanding that cancer cells exhibit increased glutamine uptake via the SLC1A5 transporter [82]. Furthermore, ^18^F-glutamine-PET/CT scans have been proposed as a possible tool to monitor the efficacy of glutamine-targeted therapies [46].

Sufficient levels of amino acids in systemic circulation are necessary to satisfy the bioenergetic needs of tumor cells in addition to providing intermediates for macromolecular synthesis [12]. Specifically, amino acids, such as glutamine, glutamate, aspartic acid, and serine are crucial for DNA synthesis, angiogenesis, and protein content amplification [83]. During the process of transformation, the increase in demand for these amino acids leads to increased consumption and subsequent lower bioavailability in cancer patients [84]. In African American and Caucasian American patients with prostate cancer, serum glutamate levels directly correlated with their Gleason score [85]. Likewise, plasma levels of glutamate are increased in colorectal carcinoma patients and in patients who have acquired immunodeficiency syndrome (AIDS) [86]. Other studies by Vanhone et al. and Rodriguez-Tomas et al. elucidate a clinical application to utilize systemic glutamate bioavailability, where they use blood plasma glutamate concentration for the diagnosis of lung cancer with higher specificity [87,88]. Interestingly, while investigating whether glutaminases function as prognostic biomarkers in human cancers, Saha et al. revealed that GLS and GLS2 expression can differentially modulate the clinical outcomes depending on the type of cancer [89]. Similar to how patients who carried the mutated BRAF genotype were found to display improved response to vemurafenib therapy [80,90], certain levels of glutamate in the blood could also provide insights into the potential responsiveness of these patients to glutamatergic inhibitors. Metabolic and signaling activities of these biomarkers could pave the way for better prognostic tools and potential therapeutic interventions.

## 9. Glutaminase and Its Inhibition

GLS is the most well-studied and also the rate-limiting enzyme in the glutaminolysis pathway. Overexpression of GLS allows for increased glutamine metabolism, thereby providing a means for the tumor cells to replenish the citric acid cycle and produce molecules required for anabolic growth. This fundamental insight afforded from basic research, which has provided to the understanding of the glutaminolysis pathway, has allowed for the development of various GLS inhibitors, such as Bis-2-(5-phenylacetamido-1,2,4-thiadazol-2yl) ethyl sulfide (BPTES), CB-839, and compound 968. These have been shown to allosterically inhibit GLS [46,91]. BPTES is specific for the kidney-type glutaminase isoform [92]. The mechanism of action of BPTES occurs by the compound binding to the dimer interface of GLS, thereby inhibiting the tetramerization of GLS, subsequently leading to its inactivation [46,91]. BPTES has also been shown to suppress cancer-cell growth in vitro and in vivo [91]. Even though BPTES is a potent inhibitor of GLS, the pharmacokinetic analysis of this compound has revealed that it has poor solubility and bioavailability, thus limiting its potential for clinical use. This led to the development of CB-839 (Telaglenastat^®^) by Calithera Biosciences [45]. CB-839, first reported by Gross et al., is a selective, noncompetitive, and potent inhibitor for GLS that has displayed antiproliferative efficacy in many cancers, including melanoma, breast cancer, leukemia/lymphoma, and kidney cancer [93,94,95]. The recent crystal structure analysis showed that the terminal electron-withdrawing trifluoromethoxy not only increases the integral lipophilicity but also improves the electronegativity of the pyridazinyl nitrogen atoms resulting in strengthened hydrogen bond interaction [96]. In particular, CB-839 is the only small molecule inhibitor of GLS that is being evaluated in clinical settings, currently in phase 1 and 2 clinical trials [97]. Additionally, another member of the GLS allosteric inhibitor family is compound 968. Compound 968 was shown to block oncogenic transformation of fibroblasts, while also displaying antiproliferative effects on cancer cells without affecting their normal counterparts [54]. The mechanism of action of compound 968 is through the binding of it to the monomeric interface of GLS, in comparison with BPTES and CB-839, which bind at the dimer interface [91]. CB-839 and BPTES are known to exclusively inhibit both products of the *GLS1* gene, GAC and KGA. However, the pan-glutaminase inhibitor compound 968 targets protein forms of both *GLS1* and *GLS2* (LGA) and has recently been utilized to suppress luminal-type breast cancer growth by inhibiting the previously underappreciated LGA [98]. In ovarian cancer cells, GLS inhibition enhances the effectiveness of chemotherapy [59] and also improves the efficacy of other targeted therapies [60,61], suggesting the critical role of targeting GLS in an attempt to improve overall patient response. Moreover, the accumulation of glutamine, as a result of GLS inhibition, has been shown to induce divergent metabolic programs to overcome tumor immune evasion [99]. This has been linked to enhanced anti-tumor activity of PD-1 and PD-L1 antibodies by overcoming the blockade of T cell activation [100]. Taken together, GLS inhibitors have shown great pre-clinical promise across cancers; however, resistance is a major hurdle of monotherapy regimes [97].

## 10. Resistance to Glutaminase Inhibition

As a monotherapy, GLS inhibition can be overcome by tumors cells through compensatory mechanisms, specifically against glutamate deprivation through different permutations of asparagine synthetase, a glutamate/cystine antiporter (xCT), or pyruvate carboxylases [101,102,103]. To overcome GLS inhibition, tumor cells have been shown to upregulate asparagine synthetase, leading to an increase in asparagine concentrations which regulates the uptake of certain amino acids, mammalian target of rapamycin complex 1 (mTORC1) activation, as well as protein and nucleotide synthesis [103]. Additionally, breast cancer cells were shown to be viable even under glucose deprivation, in conjunction with a dysfunctional xCT antiporter results in the sustenance of mitochondrial respiration [101]. It is possible that xCT expression is downregulated in CB-839-resistant cells to demote any further glutamate export. The third mechanism of resistance is the upregulation of pyruvate carboxylase [46,102]. Pyruvate carboxylase functions in the conversion of pyruvate into oxaloacetate [46,102]. In relation to glutamate-deprived cells, it can replenish the citric acid cycle and is upregulated in CB-839-resistant cancer cells [46,102]. In fact, Parlati and colleagues have suggested that pyruvate carboxylase expression strongly correlates with resistance to CB-839, and that it can rescue cells from GLS inhibition by supporting anapleurotic utilization of glucose [104]. Additionally, it is possible that the environment and metabolic milieu accompanying the tumor is responsible for the apparent resistance to glutaminase inhibition [105,106]. Looking towards the future, it might be beneficial for patients to be treated with a combinatorial drug regime that targets two or more proteins within the glutaminolysis pathway. Taking these resistant mechanisms into consideration accentuates the importance of developing a multifaceted approach towards targeting cancer-cell metabolism.

## 11. Regulation of Glutaminase

The regulation of GLS in cancer remains to be fully elucidated. Several studies have proposed different mechanisms by which GLS is regulated. Figure 2 provides a summary of how GLS is potentially regulated in GRM1+ melanoma cells. Gao et al. unfolded the indirect link between c-Myc, a well-known oncogenic transcription factor, and glutamine metabolism. c-Myc has been implicated in both activation and repression of numerous cellular functions, especially metabolism. Elevated levels of c-Myc protein transcriptionally suppress two microRNAs, miR-23a, and miR-23b, which target GLS mRNA. As a result, upregulated expression of mitochondrial GLS induces increasing amounts of glutamate and glutamate-derived metabolites into the TCA cycle to sustain neoplastic progression [62]. Liu and colleagues provided evidences on the correlation between c-Myc overexpression and the mammalian target of rapamycin (mTOR) signaling pathway, which is a critical intracellular regulator of the cell cycle. In 80% of human cancers, mTOR is abnormally activated and, thus, overstimulates many routes that the transformed cells use to synthesize proteins, lipids, and nucleotides [107]. mTOR serves as the catalytic subunits of two multi-protein complexes: mTOR complex 1 (mTORC1) and complex 2 (mTORC2). mTORC1 has been extensively studied regarding cancer-cell metabolism and has been noted as a major signaling component in regulating anabolic processes necessary for cellular growth. There is also evidence that mTORC1 mediates aerobic glycolysis via hypoxia inducible factor 1 alpha (HIF-1α), a transcription factor that functions in initiating angiogenesis and regulating cellular metabolism to overcome hypoxia [108]. The combined features of mTOR signaling have been an active topic of discussion in cancer research and one that our group has been currently investigating. Liu et al. proposed the requirement of an intact mTORC1 axis in c-Myc-driven hepatocarcinogenesis as a possible target for treatment [109]. Rapamycin, a specific inhibitor of mTORC1 activity, is useful in the treatment of certain cancers. However, studies have hypothesized that prolonged rapamycin treatment can considerably reduce levels of mTORC2 [110]. To circumvent this, everolimus was developed. Compared with the parent compound rapamycin, everolimus is more selective for the mTORC1 protein complex, with no impact on the mTORC2 complex [111]. Both rapamycin and everolimus have displayed inhibitory effects on the growth, proliferation, and survival of tumors including melanoma, with minimal toxicity [112]. AKT is a protein kinase downstream of mTORC2 and is controlled via negative feedback regulation from mTORC1. With increased inhibition of mTORC1, there may be hyper-activation of AKT, which can lead to longer cell survival in some cell types. Interestingly, it has been found that the mTORC1/c-Myc axis also regulates GLS expression in pancreatic cancer [113].

Numerous reports have uncovered alternate mechanisms underlying GLS-mediated pathogenesis. Rathore et al. discovered that NF-κB, which is initially defined as a nuclear factor that binds to the B site of the immunoglobulin κ light chain gene enhancer in B lymphocytes, exhibited similar mechanisms to switch glutamine from a non-essential amino acid to a major energy source [114]. In a human T-lymphocytic cell line, Jurkat, the p65 subunit of NF-κB binds to miR-23a and recruits the histone deacetylase (HDAC) to suppress downstream gene expression, which results in enhanced glutamine consumption [114]. Zhao et al. found that interferon-α (IFN-α) induced phosphorylation of Signal Transducer and Activator of Transcription 1 (STAT1), which then binds to a GLS promotor resulting in enhanced *GLS1* transcription [115]. Lukey et al. unveiled a vital role of the transcription factor c-Jun in metabolic reprogramming. As the product of oncogene *JUN*, c-Jun directly binds to the GLS promoter which increases gene expression in breast cancer cells [116]. Uncovering unique complex networks of GLS regulation that are specific to each cancer type introduces potential for new targeted therapeutics via a “bench to bedside” approach [117].

## 12. Questions for the Future

The high metabolic demand of cancer results in increased production of mitochondrial reactive oxygen species. To combat this, tumors increase antioxidant production via hyperactivation of the nuclear factor erythroid 2-related factor 2 (NRF2) pathway. NRF2 is the master regulator of a cell’s endogenous antioxidant response. Kelch-like ECH-associated protein 1 (Keap1) has been shown to interact with and directly promote proteasomal degradation of NRF2 by cooperating with Cul3, an important component of the E3 ubiquitin ligase complex [118]. Keap1-mutant lung cancer cells have been shown to demonstrate increased sensitivity to GLS inhibition and glutamine deprivation [119]. This sensitivity to GLS inhibition is the result of Keap1-mutated cells being overly dependent on glutaminolysis through proper functioning of xCT transporter [120]. Pharmacologic modulation of the NFR2/GSH pathway paired with subsequent alterations in the expression of xCT downstream could serve as a predictor of cellular response to resistance and/or sensitivity to certain drugs [121,122]. Moreover, high expression levels of genes related to GSH synthesis, such as glutamate cysteine ligase (GCL) have been shown to promote resistance to anti-cancer treatments [123]. These findings could provide additional insight into the involvement of glutamate utilization. A better understanding of how the NRF2/Keap1 axis functions at the molecular level and how it connects to the glutamatergic pathway in melanoma may help uncover novel regulatory mechanisms of GLS-mediated tumorigenesis.

Accompanied by the onset of the post-genome era, scientists are now beginning to divert their attention from conventional “one-size-fits-all” therapy to personalized medicine. As the first one to discover that ectopic expression of GRM1 is the driving basis for melanoma development, our group has been actively investigating glutamatergic signaling inhibitors to treat melanoma in experimental models and patients with limited success. Out future goal is to combine inhibitors targeting distinctive but complementary glutamatergic signaling pathways for the treatment of melanoma. A promising candidate as the complementing inhibitor is CB-839. Currently, CB-839 is in clinical trials in combination with other compounds for patients with advanced, metastatic, solid, and hematopoietic tumors [124,125], where glutamine metabolism has been identified as a suitable drug target. Glutaminase inhibition has also been postulated to prime the immune system and improve patients’ responsiveness to immune checkpoint therapy [126,127]. We hope to fully identify and unveil the most efficient combination therapy targeting glutamatergic signaling; current therapies may be optimized to prolong the survival of patients.

## 13. Conclusions

Mounting data on aberrant metabolic pathways in cancer etiology suggest deregulated activities of some of the key enzymes constituting specific metabolic pathways could be significant contributors to cancer development and progression. One of the most noticeable reprogramming events in cancer cell metabolisms is the preferential reductive glutamine metabolism even in normoxic conditions. Glutamine, the most abundant circulating amino acid in human plasma, provides considerable nutrient sources including carbons to highly proliferative tumor cells for the production of TCA cycle intermediates, fatty acids, nucleotides and nonessential amino acids. The vital role of glutamine metabolism in cancer cell proliferation suggests glutaminolytic enzymes could be attractive targets for therapy. GLS (glutaminase) converts glutamine to glutamate; elevated GLS levels have been found in tumors and are functionally linked to several oncogenic transcription factors including Myc, NF-kB, c-Jun-induced cell growth in some cancers suggesting the potential of GLS as an important player in therapeutic strategy.

## Figures and Tables

**Figure 1 cancers-12-02624-f001:**
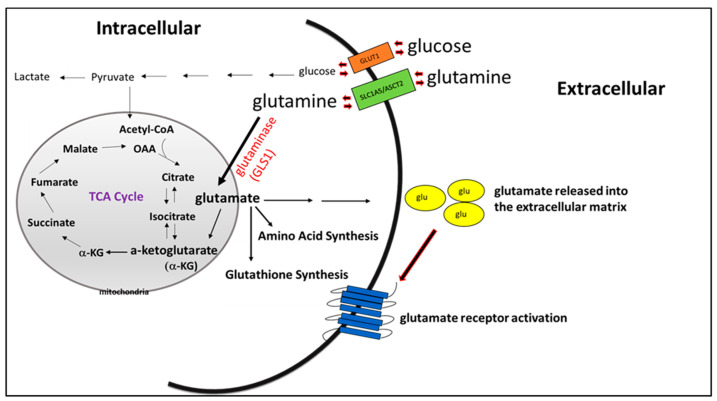
Tumor cells utilize both glutamine and glucose for growth and energy production. Here, we show the several fates of glutamate produced as a result of glutaminolysis.

**Figure 2 cancers-12-02624-f002:**
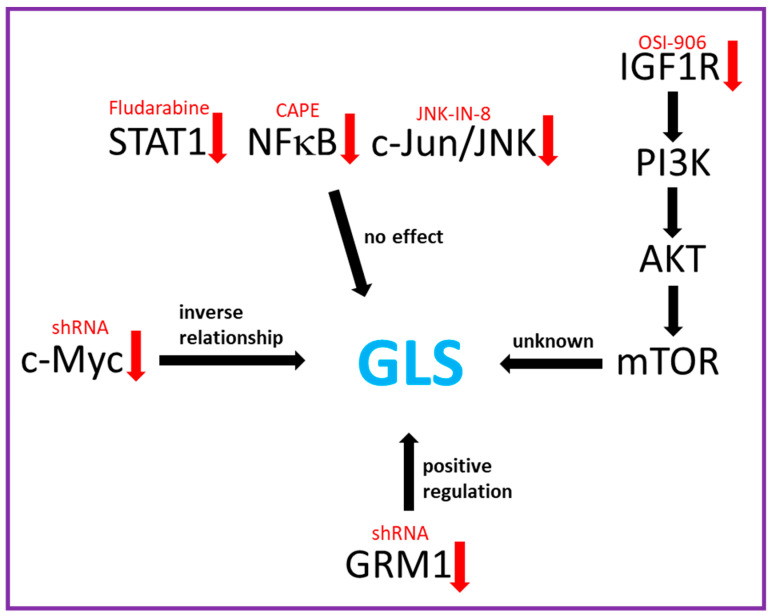
A summary of the proposed pathways/proteins responsible for the regulation of GLS in GRM1^+^ melanoma. Red-colored arrows indicate inhibition and black-colored arrows indicate regulation. Red font indicates the mode of inhibition.

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
