# Peer review of "Metabolic Signaling Cascades Prompted by Glutaminolysis in Cancer"

_cancers, 2020, doi:10.3390/cancers12092624_

Round 1
Reviewer 1 Report
The review submitted by Shah and Chen provides a brief summary on metabolic fate of glutamine in cancer cells, with the stress put on glutaminase.
The first four paragraphs are general but original references are required, aside the reviews.
The first sentence of abstract informs that the melanoma cells will be in focus of the review but this is not the case, however. This requires some amendments – at least original articles illustrating the described processes by melanoma examples. My general advice would be to focus on melanoma cells and greatly extend this aspect, however.
The paragraph 5 could be renamed and moved after the paragraph 7, to preserve the logic: glutaminolysis, glutamate synthesis and release (augmented in cancer), and what the cancer use the glutamate for. If keeping the present sequence or not, the physiological role of glutamate should be described in more detail, especially the signalling but also transport.
Nevertheless, also the role of glutamate in cancer cells should be elaborated (molecular pathways, signalling, etc.).
Regulation of glutaminase: is just one of provided references pertains direct relationship between c-myc and glutaminase? Are there any other research available on that?
In terms of the role of glutaminase role in cancer, I encourage to consider including works of Javier Marquez and Monika Szeliga on glutaminases in gliomas (for references see e.g. DOI: 10.3390/cancers11010115).
Author Response
Reviewer 1:
The review submitted by Shah and Chen provides a brief summary on metabolic fate of glutamine in cancer cells, with the stress put on glutaminase.
The first four paragraphs are general but original references are required, aside the reviews. The first sentence of abstract informs that the melanoma cells will be in focus of the review but this is not the case, however. This requires some amendments – at least original articles illustrating the described processes by melanoma examples. My general advice would be to focus on melanoma cells and greatly extend this aspect, however.
Thank you for pointing out this discrepancy. We have removed the introductory sentence that mistakenly states the exclusive focus of this review on melanoma. Using a melanoma model system, our laboratory has recently published a research article illustrating the link between glutaminolysis and glutamatergic signaling. Findings from this article encouraged us to share this review that utilizes examples of various cancers with a focus on the involvement of glutaminolysis in association with metabolic signaling.
The paragraph 5 could be renamed and moved after the paragraph 7, to preserve the logic: glutaminolysis, glutamate synthesis and release (augmented in cancer), and what the cancer use the glutamate for. If keeping the present sequence or not, the physiological role of glutamate should be described in more detail, especially the signalling but also transport.
Nevertheless, also the role of glutamate in cancer cells should be elaborated (molecular pathways, signalling, etc.).
We appreciate your suggestion, but after carefully reviewing the sequences of the topic, we felt that introducing glutamate before delving into associated details helps with the readability of this section. However, we have elaborated on the physiological roles of glutamate in both normal and cancer cells.
Regulation of glutaminase: is just one of provided references pertains direct relationship between c-myc and glutaminase? Are there any other research available on that?
In terms of the role of glutaminase role in cancer, I encourage to consider including works of Javier Marquez and Monika Szeliga on glutaminases in gliomas (for references see e.g. DOI: 10.3390/cancers11010115).
Regarding the question pertaining to the direct relationship between c-Myc and GLS1, there is only one study that elucidates this clearly. There are indeed other studies that highlight a direct link between c-Myc and GLS2 (PMID: 26528759). GLS2 is not the focus of this manuscript. As per your suggestion, we did look into the works of Javier Marquez and Monika Szeliga – they illustrated the involvement of GLS2 in sensitizing human glioblastoma via inhibition of the MAP kinase pathway. Considering GLS1 not GLS2 is the focus of this review, we decided not to include the research studies on GLS2.
Reviewer 2 Report
In the manuscript titled “Metabolic signaling cascades prompted by glutaminolysis in cancer” prepared by Shah and Chen, the author reviewed the essential concepts in glutamate metabolism in cancer. I think this is a well-prepared manuscript which covers the critical points in the glutamate metabolism in cancer. I have several minor concerns as follow:
- The author introduced aerobic glycolysis and Warburg effect in section 2. It gives me the impression that Warburg metabolism is a consequence of hypoxic conditions. However, in several previous articles (e.g. PMID: 18775299), it has been shown that cancer cells prefer to use Warburg metabolism, even in the context of oxidative environment. Based on another review (PMID: 25689954), the Warburg metabolism could be initiated by HIF-alpha or PI3K/Akt/mTOR pathways, suggesting that cancer-associated molecular signaling proactively utilize glycolysis for their metabolic demands.
- In section 12, the author mentioned the NRF2-derived xCT expression is identified in Keap1 mutated cells. It is also essential to mention that several recent studies (e.g., PMID: 16103098, PMID: 31548295) highlight the role of other NRF2 downstream, such as glutamate-cysteine ligase (GCL), to utilize glutamate for glutathione metabolism. The enhanced glutathione metabolism could serve as a metabolic support for cancer cells.
- It would be helpful for the author to compose a schematic illustration, highlighting the central role of glutamate metabolism among cancer metabolism.
Author Response
Reviewer-2:
In the manuscript titled “Metabolic signaling cascades prompted by glutaminolysis in cancer” prepared by Shah and Chen, the author reviewed the essential concepts in glutamate metabolism in cancer. I think this is a well-prepared manuscript which covers the critical points in the glutamate metabolism in cancer. I have several minor concerns as follow:
- The author introduced aerobic glycolysis and Warburg effect in section 2. It gives me the impression that Warburg metabolism is a consequence of hypoxic conditions. However, in several previous articles (e.g. PMID: 18775299), it has been shown that cancer cells prefer to use Warburg metabolism, even in the context of oxidative environment. Based on another review (PMID: 25689954), the Warburg metabolism could be initiated by HIF-alpha or PI3K/Akt/mTOR pathways, suggesting that cancer-associated molecular signaling proactively utilize glycolysis for their metabolic demands.
Thank you for point this out. We have now rectified the relevant sentence to correct for the misconceived notion. Additionally, we have added findings from the aforementioned review (PMID: 25689954) to this section.
- In section 12, the author mentioned the NRF2-derived xCT expression is identified in Keap1 mutated cells. It is also essential to mention that several recent studies (e.g., PMID: 16103098, PMID: 31548295) highlight the role of other NRF2 downstream, such as glutamate-cysteine ligase (GCL), to utilize glutamate for glutathione metabolism. The enhanced glutathione metabolism could serve as a metabolic support for cancer cells.
We appreciate your suggestion. We have now included relevant findings from these articles into this section.
- It would be helpful for the author to compose a schematic illustration, highlighting the central role of glutamate metabolism among cancer metabolism.
Thank you for this fantastic suggestion. We have incorporated this figure into the revised manuscript.
Reviewer 3 Report
This is an interesting review that highlights two related features of cancer physiology: glutamine metabolism and glutamate signaling and metabolism. The dependence of cancer cells on glutamine metabolism is widely acknowledged and was/is a subject of numerous high-profile reviews by Chi Dang and others. On the other hand, the senior author of this manuscript, Suzie Cheng, and few other laboratories discovered that the product of glutamine degradation, glutamate, can promote growth and progression of several types of cancers, perhaps via mGluR signaling. This latter aspect of tumor biology is less appreciated and for this reason represents an attractive aspect of the present paper. Unfortunately, I think that the “glutamate” part of the story is (a) underdeveloped and (b) in several instances overinterpreted in relationship to what is known in the field. There is a room for improvement here.
My specific comments are enumerated below.
[1] Lns. 51-53: It is erroneous to state that glycolysis can generate more ATP that oxidative phosphorylation due to its faster rate. The authors of the quoted manuscript [25] actually suggest that glycolysis provides an advantage for cell growth under conditions of limited glucose supply. No relevance to higher/lower ATP levels is considered.
[2] Section 5 on “normal” roles for glutamate is a bit superficial. I would NOT start with glutamate signaling in the CNS, which is evolutionary secondary to the metabolic roles for this amino acid. The section misses the essential role of this amino acid in nitrogen transfer, both in the anabolic and catabolic reactions of transamination. It is for the latter reason, glutamate is (one of) the most abundant amino acid in cytosol and mitochondrial matrix. Glutamine transferases deserve some attention too for their role in normal physiology and cancers.
[3] Lns. 101-102: The authors overrepresent conclusions of paper [25] on the role of glutamate in tumor progression. The study [25] actually demonstrated that glutamate and glutamate receptor agonists reduce tumor cell proliferation.
[4] Lns. 102-103: To the best of my knowledge, study [26] established very limited effects of glutamate on growth of SOME melanoma cell lines and no evidence for receptor signaling was provided.
[5] Lns. 104-106: study [27] is misquoted because it was focused on glutamine rather than glutamate.
[6] Section 7 can be extended. Besides the Chen group work on mGluR1, there were numerous other studies on other mGluRs and, most importantly, iGluRs. I think that review of Stepulak et al. in J Neural Transm (2014) does a good job of covering many relevant topics. Transcellular signaling in glioblastoma tumors may be further expanded based on numerous reviews of Sontheimer group. Perhaps, I my memory fails me, but there were a few manuscripts suggesting the role of glutamate release and transcellular glutamate signaling in bone metastasis too.
Author Response
Reviewer-3:
This is an interesting review that highlights two related features of cancer physiology: glutamine metabolism and glutamate signaling and metabolism. The dependence of cancer cells on glutamine metabolism is widely acknowledged and was/is a subject of numerous high-profile reviews by Chi Dang and others. On the other hand, the senior author of this manuscript, Suzie Cheng, and few other laboratories discovered that the product of glutamine degradation, glutamate, can promote growth and progression of several types of cancers, perhaps via mGluR signaling. This latter aspect of tumor biology is less appreciated and for this reason represents an attractive aspect of the present paper. Unfortunately, I think that the “glutamate” part of the story is (a) underdeveloped and (b) in several instances overinterpreted in relationship to what is known in the field. There is a room for improvement here.
My specific comments are enumerated below.
[1] Lns. 51-53: It is erroneous to state that glycolysis can generate more ATP that oxidative phosphorylation due to its faster rate. The authors of the quoted manuscript [25] actually suggest that glycolysis provides an advantage for cell growth under conditions of limited glucose supply. No relevance to higher/lower ATP levels is considered.
Thank you for highlighting this error. We have rephrased this sentence in the revised manuscript.
[2] Section 5 on “normal” roles for glutamate is a bit superficial. I would NOT start with glutamate signaling in the CNS, which is evolutionary secondary to the metabolic roles for this amino acid. The section misses the essential role of this amino acid in nitrogen transfer, both in the anabolic and catabolic reactions of transamination. It is for the latter reason, glutamate is (one of) the most abundant amino acid in cytosol and mitochondrial matrix. Glutamine transferases deserve some attention too for their role in normal physiology and cancers.
As per your suggestion, we have added a section describing the essential role of glutamate in nitrogen transfer.
[3] Lns. 101-102: The authors overrepresent conclusions of paper [25] on the role of glutamate in tumor progression. The study [25] actually demonstrated that glutamate and glutamate receptor agonists reduce tumor cell proliferation.
If glutamate is involved in tumor progression, utilizing glutamate and glutamate receptor antagonists would lead to reduced tumor growth and progression.
[4] Lns. 102-103: To the best of my knowledge, study [26] established very limited effects of glutamate on growth of SOME melanoma cell lines and no evidence for receptor signaling was provided.
Glutamate is an essential amino acid that is required for the growth and survival of all cells as it maintains energy homeostasis. Several studies have shown that some melanoma cells are dependent on both glutamate (ligand) and glutamate receptor signaling to maintain the transformed phenotype. To emphasize this, we have cited a couple of other studies in the revised manuscript.
[5] Lns. 104-106: study [27] is misquoted because it was focused on glutamine rather than glutamate.
Thank you for pointing this out. We have made the rectification and replaced the erroneous reference with the correct one.
[6] Section 7 can be extended. Besides the Chen group work on mGluR1, there were numerous other studies on other mGluRs and, most importantly, iGluRs. I think that review of Stepulak et al. in J Neural Transm (2014) does a good job of covering many relevant topics. Transcellular signaling in glioblastoma tumors may be further expanded based on numerous reviews of Sontheimer group. Perhaps, I my memory fails me, but there were a few manuscripts suggesting the role of glutamate release and transcellular glutamate signaling in bone metastasis
We appreciate your suggestion. A section on the involvement of ionotropic glutamate receptors has now been included in Section 7 of the revised manuscript.
Round 2
Reviewer 3 Report
I am satisfied with the Authors' responses.